# Development and Functionality of Sinami (*Oenocarpus mapora*) Seed Powder as a Biobased Ingredient for the Production of Cosmetic Products

Frank L. Romero-Orejon [1,*], Jorge Huaman [2], Patricia Lozada [2], Fernando Ramos-Escudero [1,3] and Ana María Muñoz [4]

1   Unidad de Investigación en Nutrición, Salud, Alimentos Funcionales y Nutraceúticos, Universidad San Ignacio de Loyola (UNUSAN-USIL), Calle Toulon 310, Lima 15024, Peru; dramos@usil.edu.pe
2   Centro de Investigación, Tecnología e Innovación Cosmética, Universidad San Ignacio de Loyola (CITIC-USIL), Av. La Fontana 550, Lima 15024, Peru; jhuaman@usil.edu.pe (J.H.); jlozada@usil.edu.pe (P.L.)
3   Facultad de Ciencias de la Salud, Universidad San Ignacio de Loyola, Av. La Fontana 750, Lima 15024, Peru
4   Instituto de Ciencias de los Alimentos y Nutrición, Universidad San Ignacio de Loyola (ICAN-USIL), Campus Pachacamac, Lima 15823, Peru; amunoz@usil.edu.pe
*   Correspondence: fromero@usil.edu.pe; Tel.: +51-940206594

**Abstract:** Sinami (*Oenocarpus mapora* H. Karst), a native fruit found in the Amazon region of South America, has high nutritional value and is rich in lipids. However, the processing of sinami generates a large volume of agro-industrial waste, mainly composed of seeds. Our research comprises a proximal analysis of the sinami seed and its phenolic compounds and their antioxidant activity. The chemical analysis revealed high moisture but low protein, fiber, and lipid content. Furthermore, the extracts showed high in vitro antioxidant activity against 1,1-diphenyl-2-picrylhydrazyl radical $IC_{50}$ ($0.34 \pm 0.001$ mg/mL) and ABTS IC50 ($0.10 \pm 0.0002$ mg/mL) free radicals. Based on this previous assessment, a gel exfoliant was developed. Since sinami seed powder is a novel ingredient, different formulations were evaluated to determine future incorporation into the cosmetic market. The best exfoliant gel prototype was studied under normal and stressed conditions (40 °C) for 3 months, maintaining a pH value of 5.25 and final viscosities of 700–800 mPa.s and 600–500 mPa.s under normal and stress conditions, respectively. Although unexplored, the sinami seed could be considered a raw material for the cosmetic industry.

**Keywords:** *Oenocarpus mapora* H. Karst; formulation; body scrub; cosmetics; revalorization; physical stability

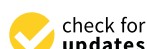



## 1. Introduction

The family Arecaceae, which includes more than 150 species, is of great importance to the traditional populations and indigenous communities of the Amazon region [1]. Their biodiversity has spurred interest in the use of native and exotic palm trees and research in favor of developing and designing better agrotechnological products [2]. Palm trees are variously used for human consumption, utensils, tools, construction, and industrial purposes, including pharmaceuticals, chemicals, dermocosmetics, oleochemicals, and biofuels [3–5].

Palms of the genus *Oenocarpus*—a tree belonging to the botanical family Arecaceae—can be found from the floodplain forests of southern Costa Rica to the Maracaibo basin in Venezuela, in addition to the western and central parts of the Amazon basin [6]. The genus *Oenocarpus* includes different species, such as *O. bataua*, *O. bacaba*, *O. distichis*, and *O. mapora*. Fruits of this genus are elliptical to globose in shape and dark purple when ripe [7].

Sinami (*O. mapora*)—also called "bacabinha," "mil pesillos," or "sinamillo" in South American countries—is consumed by the local population as pulp or wine, similar to açaí fruit (*Euterpe oleracea*). It is also used as a raw material in producing ice creams, popsicles, jellies, and liqueurs [8]. In Brazil, the oil is extracted from the mesocarp for cooking and used as a skin conditioner [6]. This oil is rich in oleic acid and resembles olive oil [9,10] and is rich in vegetable pigments—including chlorophyll and carotenoids [11].

Sinami pulp and peel supply polyphenols, fatty acids, and antioxidant activity and are therefore used for a variety of purposes [12]. However, there are no reported uses for the seeds. In their investigation, Ferrer Cutire et al. [13] found that the residual sinami cake has a high content of carbohydrates (76.0 g/100 g) and crude fiber (19.20 g/100 g). The lipid profile revealed multiple fatty acids, including oleic (60.61% $\pm$ 0.10%), palmitic (20.19% $\pm$ 0.11%), and linoleic (12.99% $\pm$ 0.11%) varieties. Highly established oils, such as those obtained from the seeds of *passiflora edulis*, can provide emollient action, thus increasing the skin's hydration and elasticity [14].

Since it can be used as a biodegradable exfoliating ingredient, sinami could be an ecological alternative to cosmetic products. Cosmetic exfoliation improves the skin's appearance by removing dead cells [15] using one of two methods. Chemical exfoliation uses organic acids ($\alpha$-hydroxy acid), enzymes (papain and bromelin), or other active chemicals that interact with the skin. In contrast, physical exfoliation involves rubbing small abrasive particles against the skin. These particles could take the form of synthetic polymers or fine seed powder [16]. However, the cosmetic use of sinami seed largely depends on its compatibility, as a physical exfoliant, with currently used cosmetic ingredients.

Valorization of agro-industrial waste allows the recovery of value-added components. Sinami seed could be incorporated into cosmetics as a chemical or physical exfoliant. The seeds are supplied by Amazonian communities that supply the local and regional markets with sinami fruit; these populations reside near areas where palm trees flourish or have productive groves [7]. Thus, reexamining our use of this by-product—the sinami seed—is necessary to understand the economic effects on local producers or companies that work exclusively with the fruit's pulp.

This work fills a critical gap in the literature regarding *O. mapora*. We analyze its proximal composition, evaluate its phenolic compounds, and examine its antioxidant activity using two different assays. The first, a gel-type cosmetic prototype incorporated sinami seed powder as an exfoliant to determine its potential for future use in the cosmetic industry. The best product prototypes underwent 3 months of stability testing after formulation under stress conditions.

## 2. Materials and Methods

### 2.1. Chemical Products

The following standards and reagents were used: 2,2′-azino-bis (3-ethylbenzothiazoline-6-sulfonic acid (ABTS), gallic acid, sulfuric acid, sodium carbonate, ethanol, 1,1-diphenyl-2-picrylhydrazyl radical (DPPH), hexane, sodium hydroxide, and Folin–Ciocalteu reagent. All were purchased from Sigma-Aldrich.

The following inputs were used to formulate the exfoliant prototype: Acrylates/C10–30, cocamidopropyl betaine, disodium cocoyl glutamate, glycerin, hydroxyacetophenone, fragrance, sodium gluconate, polyquaternium-7, triethanolamine, sodium laureth sulfate, and stearyl behenate. The chemicals were obtained from Insuquimica (Lima, Perú).

### 2.2. Sampling and Treatment of Seeds

We selected only sinami seeds without signs of deterioration or physical damage. The sample was manually reduced and dried for 4 h (UF160, Memmert, Schwabach, Germany). Finally, the sample size was reduced with a laboratory mill at 150 RPM for 10 s (GRINDOMIX GM 200, Retsch, Düsseldorf, Germany). The samples were vacuum packed (SU-316, Sammic, IL, USA) and stored at $-20$ °C (A54546, Infraca, Valencia, Spain) until

the proximal analysis, phenolic content and antioxidant tests were performed. Figure 1 shows the flowchart for obtaining sinami seed powder.

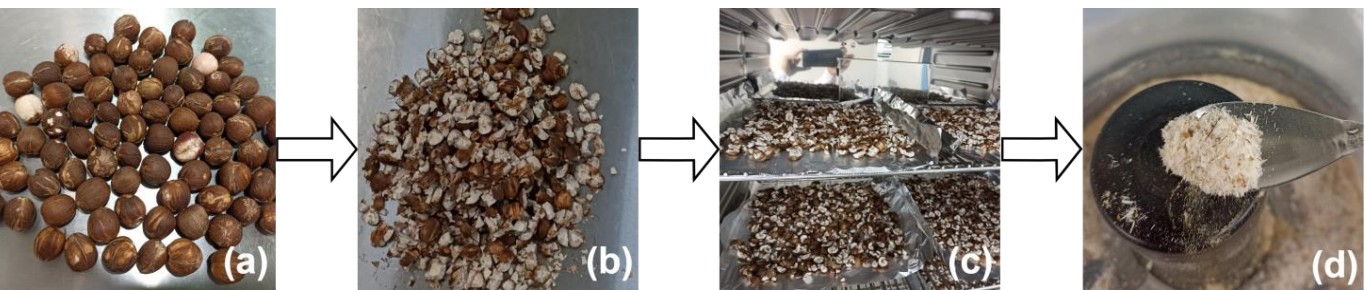

**Figure 1.** Summary of the steps employed to obtain the sinami seed powder from fresh seeds. (**a**) Fresh shelled seeds; (**b**) manually reduced seeds; (**c**) drying; (**d**) mechanically reduced seeds.

Obtaining the sinami seed powder as an exfoliant agent required sieving. The powder sample was separated by a sieve according to particle size using six test sieves (AS 200 control, Retsch, Düsseldorf, Germany). The sample was placed on the top sieve, and the sieving equipment was vibrated for 15 min per batch. The Retsch test sieves used were 20 (850 μm), 35 (500 μm), 60 (250 μm), 80 (180 μm), and 120 (125 μm) size. The process of sieving the *Oenocarpus mapora* seeds is shown in Figure 2.

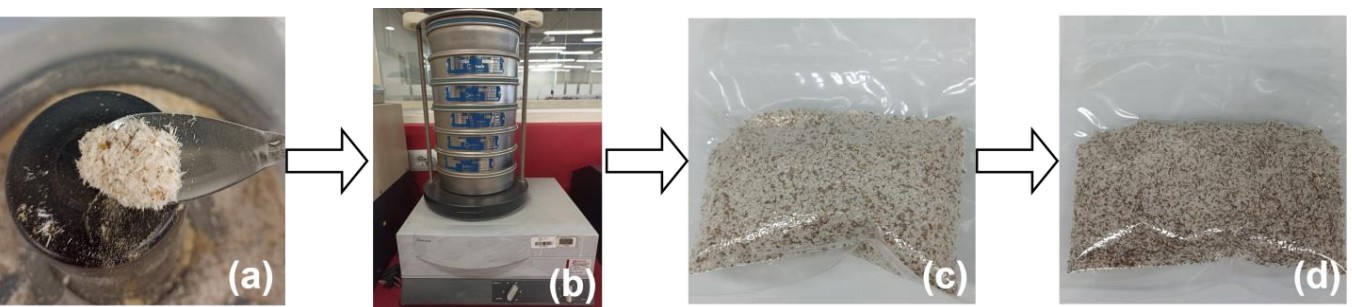

**Figure 2.** Sinami seed powder sieving flowchart. (**a**) Sinami seed powder; (**b**) sieve; (**c**) sample sieved > 500 μm; (**d**) sample sieved > 250 μm.

### 2.3. Physicochemical Characterization of the Sinami Seeds
2.3.1. Proximal Composition

The chemical composition of the sinami seed powder was determined according to the procedures established by the Association of Official Analytical Chemists (AOAC). Moisture content was measured with the AOAC 930.15 methodology AOAC [17], (AOAC, 2000) using a forced air oven (UF160, Memmert, Schwabach, Germany). Lipids were extracted with hexane using the Soxhlet method and the Extraction Unit (E-816 SOX, BÜCHI, Flawil, Switzerland). Protein content was determined using a KJELDATHERM® digester (KT-L 8s, Gerhardt, Königswinter, Germany) coupled with the VAPODEST® (500, Gerhardt, Königswinter, Germany) for distillation and titration. A Lindberg/Blue M Moldatherm™ box furnace (BF51842C-1, Thermo Scientific, Massachusetts, USA) at 600 °C was used to determine the ash content (AOAC 935.09). To determine the total crude fiber content, the methodology described by Ferrer Cutire et al. [13] was applied with some modifications, while total carbohydrates were determined by difference.

2.3.2. Extraction Process

The polar fraction was determined using 1 g of sinami seed powder, extracted in 10 mL of 80% ethanol solution. The samples were constantly shaken for 30 min in total darkness (Thermo Scientific, Waltham, MA, USA). Then, an ultrasonic bath (160 W) was used at

40 KHz and 40 °C for 30 min (CPX5800H-E, Bransonic, Danbury, CT, USA). Next, the samples were transferred to 50 mL polypropylene centrifuge tubes and centrifuged at 3500 rpm and 25 °C for 15 min (5810R, Eppendorf, Hamburg, Germany). Finally, the supernatant was recovered and stored at 4 °C prior to polyphenol and antioxidant content analyses.

### 2.3.3. Determination of Total Phenolic Content

The total polyphenol content analysis was conducted using a modification of Ferrer Cutire et al.'s method [13]. Briefly, 10 μL aliquots of sinami seed extract were added to 90 μL of purified water. Then, 750 μL of 0.2-N Folin–Ciocalteu reagent was added. After 5 min, 750 μL of 7.5% sodium carbonate was added to each microtube. After a 2 h incubation under dark conditions, absorbance at 765 nm was measured using a spectrophotometer (Jasco spectrophotometer, V-770, Jasco, Tokyo, Japan). The results are expressed in milligrams of gallic acid equivalents (GAE) per kg of sinami seed (mg GAE/kg).

### 2.4. Antioxidant Activity
### 2.4.1. DPPH Radical Scavenging Activity

The antioxidant activity was measured using the DPPH assay according to the method described by Muñoz et al. [11]. We incubated 50 μL aliquots of sinami seed extract with 950 μL of 100 μmol/L DPPH (in 60% ethanol). The samples were then shaken constantly for 30 min (Thermo Scientific, Waltham, MA, USA). Absorbance was measured at 515 nm (Jasco spectrophotometer, V-770, Jasco, Tokyo, Japan), and the results are expressed as 50% inhibitory concentrations ($IC_{50}$-DPPH).

### 2.4.2. ABTS Radical Scavenging Activity

The free radical scavenging capacity of ABTS was measured according to Melo et al.'s method [18]. Here 100 μL sample aliquots of sinami extract were incubated with 900 μL of ABTS moiety. After 30 min of reaction under dark conditions, absorbance was measured at 734 nm. The results are expressed as 50% inhibitory concentrations ($IC_{50}$-ABTS).

### 2.5. Exfoliant Formulation

Cosmetic products are mostly developed using a trial-and-error approach wherein experiments are used to determine formulations that converge with the stated objectives [19]. This investigation starts with 12 possible ingredients, including emollients, thickeners, and surfactants. Table 1 shows the proposed elements.

**Table 1.** Potential cosmetic ingredients and their functions.

| Functionality | Ingredient | INCI Name |
|---|---|---|
| Emulsion stabilizing and viscosity controlling | Carbopol ultrez 20 | Acrylates/C10–30 |
| Surfactant—emulsifying | Alkopon | Sodium laureth sulfate |
| Surfactant—emollient | Cocamidopropyl | Cocamidopropyl betaine |
| Surfactant—emulsifying | Lathanol powder | Sodium lauryl sulfoacetate |
| Surfactant—emulsifying | Hostapon sci 85 p | Sodium cocoyl glycinate |
| Humectant and solvent | Glycerin | Glycerin |
| Preservative | Procide cg | Methylchloroisothiazolinone |
| Viscosity controlling | Table salt | Sodium chloride |
| Preservative | Poliquaternium-7 | Poliquaternium-7 |
| pH control | Triethanolamine | triethanolamine |

We determined multiple feasible formulations of a gel exfoliant that used sinami seed powder using heuristic rules and tests to determine the ideal amounts and combinations of ingredients. The different formulations presented drawbacks, but the objective was reached after several trials. Table 2 shows the corresponding final gel exfoliant formulation.

**Table 2.** Final formulation used in this research and ingredient's proportions.

| Phase | N° | Ingredient | INCI Name | Functionality | % |
|---|---|---|---|---|---|
| A | 001 | Aqua | Aqua | Driver | 81.97 |
| | 002 | Carbopol Ultrez 20 | Acrylates/C10–30 | Viscosity controlling | 1.00 |
| B | 003 | Alkopon | Sodium laureth sulfate | Surfactant—emulsifying | 7.00 |
| | 004 | Cocamidopropyl Betaina | Cocamidopropyl Betaina | Surfactant—emollient | 1.50 |
| C | 005 | Glycerin | Glycerin | Humectant and solvent | 2.00 |
| | 006 | Prodice CG | Methylchloroisothiazolinone | Preservative | 0.10 |
| | 007 | Vanilla essence | - | Masking agent | 0.30 |
| D | 008 | Red dye | CI 75470- | Cosmetic colorant | 0.18 |
| | 009 | Yellow dye | CI 19140 | Cosmetic colorant | 0.12 |
| | 010 | Brilliant Blue dye | CI 42090- | Cosmetic colorant | 0.03 |
| | 011 | Triethanolamine | Triethanolamine | pH control | 0.80 |
| | 012 | Sinami seed powder | - | Abrasive agent | 5.00 |

The base formula of the gel exfoliant was prepared according to the following process. Half of the water was heated in a beaker up to 70 °C, and the acrylates/C10–30 were added; dispersion was carried out with constant stirring at 2500 rpm with an overhead stirrer (HT-120AX, Witeg, Wertheim, Germany). This mixture is called "phase A." Ingredients 003 and 004 were weighed separately and mixed with the rest of the water (phase B) in a homogenizer for 10 min (HG-15D, Witeg, Wertheim, Germany). Phase B was subsequently incorporated into phase A. Then, ingredients 005 to 007 were added to the mixture (phase C). With the help of a palette, phase C was finally mixed, and the artificial colors (008–010) were added. The sinami seed powder was added at 5% of the total weight (same as the commercial product of Euterpe oleracea used as reference), as well as triethanolamine to achieve the required pH value. The formulation was mixed until homogenized. Subsequently, duplicate 150 g samples of exfoliant gel were transferred to glass jars for stability testing.

### 2.5.1. pH and Texture

Each prototype's pH was measured directly, in triplicate, using a potentiometer (Orion Star A211, Thermo Scientific, Waltham, MA, USA). Viscosity was determined in duplicate using a viscometer (DV-E, Brookfield, Middleboro, MA, USA). We used 120 mL of each sample, and duplicate viscosity measurements were performed at 20 °C.

### 2.5.2. Storage Stability Test

The prototype with better conditions underwent preliminary stability testing comprised of two trials: (1) direct exposure to sunlight at room temperature and (2) temperature variation over time to assess physical stability under stress conditions. All tests were conducted in duplicate over the three-month evaluation period. The samples from test 1 were stored in the laboratory, exposing them directly to the sun, night and day, at room temperature (20 °C). For trial 2, the samples were stored in an oven at 40 °C (UF160, Memmert, Schwabach, Germany). During the stability tests, pH and viscosity measurements were made after the samples were allowed to reach room (20 °C) temperature.

## 3. Results

### 3.1. Characterization of the Sinami Seed

The results of the proximal analysis, humidity, and total fiber of the sinami seed powder are summarized in Table 3. The table is complemented with the proximal composition data of the seed of the açaí fruit (*Euterpe oleracea*). Both belong to the same family (Arecaceae) and *Euterpe oleracea* has been the focus of the most current studies [18], for this reason it is used as a reference in the discussion.

**Table 3.** Physicochemical characterization of *Oenocarpus mapora* and *Euterpe oleracea* seeds.

| Parameter [1] | *Oenocarpus mapora* | *Euterpe oleracea* (Reference) |
|---|---|---|
| Moisture (%) [3] | 25.1 ± 0.19 | 7.91 ± 0.01 |
| Lipids (%) [3] | 0.44 ± 0.03 | 2.75 ± 0.01 |
| Crude protein (N × 5.3) (%) [3] | 3.43 ± 0.05 | 4.89 ± 0.03 |
| Ash (%) [2,3] | 1.01 ± 0.01 | 1.36 ± 0.01 |
| Total fiber (%) [3] | 9.94 ± 0.46 | - |

[1] Data are given on a humid base; [2] data are given on a dry base; [3] all data are the average of 2 determinations ± SD.

### 3.2. Characterization of Total Phenolic Content and Antioxidant Activity

The sinami seed extracts were prepared following the conditions established in the experimental design. Ethanol was chosen due to its good extractability, wide availability, and lower cost than other organic solvents [18]. The sinami seed extracts demonstrated a total phenolic content of 12.3 mg/g (Table 4).

**Table 4.** Total phenolic content (TPC) and $IC_{50}$ values of sinami seed (*Oenocarpus mapora* H. Karst) extract.

| Sample | TPC (mg/g) | DPPH $IC_{50}$ (mg/mL) | ABTS $IC_{50}$ (mg/mL) |
|---|---|---|---|
| *Oenocarpus mapora* | 12.3 ± 0.25 | 0.34 ± 0.001 | 0.10 ± 0.0002 |

We tested two antioxidant mechanisms of action in sinami seed extracts (DPPH and ABTS). The antioxidant activity of the sinami sample ($IC_{50}$) was 0.34 µg/mL and 0.1 µg/mL, respectively (Table 4). The DPPH and ABTS radical inhibition kinetics curves for various concentrations are shown in Figure 3.

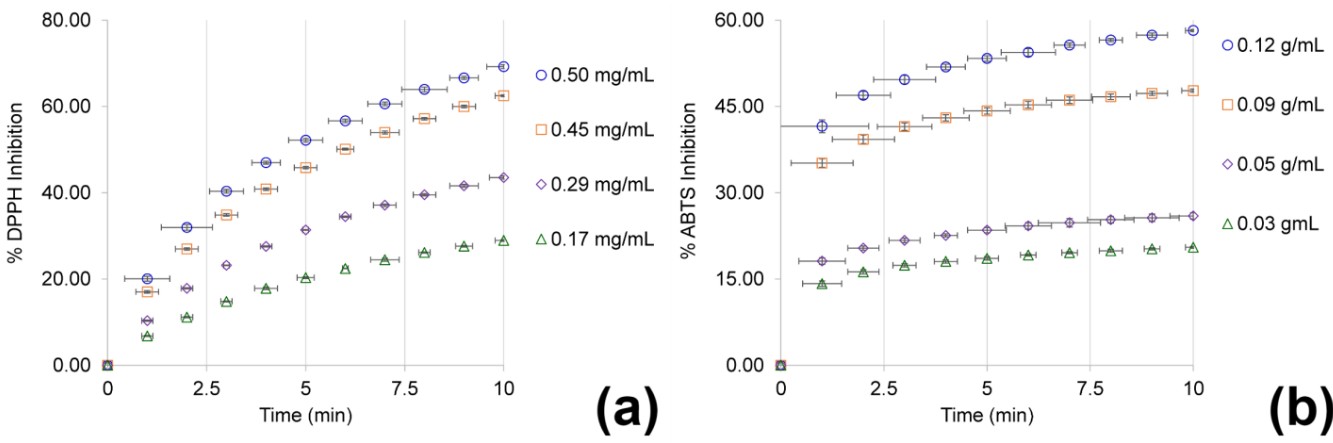

**Figure 3.** Antioxidant activity ($IC_{50}$) of different concentrations of the extract of sinami seeds (*Oenocarpus mapora* H. Karst) over 10 min. (**a**) Kinetic curves of DPPH percent inhibition; (**b**) kinetic curves of ABTS percent inhibition.

### 3.3. Exfoliant Prototype Characterization

Sinami-seed-based gel exfoliants were characterized according to particle size. The sizes of the most widely used abrasive agents are 250–500 µm, according to Azconia [20]. We decided to continue the experiment with the smaller powder size. Figure 4 shows the granular particles of 250 µm that were chosen by stereo microscope (NexiusZoom, Euromex, Arnhem, The Netherlands).

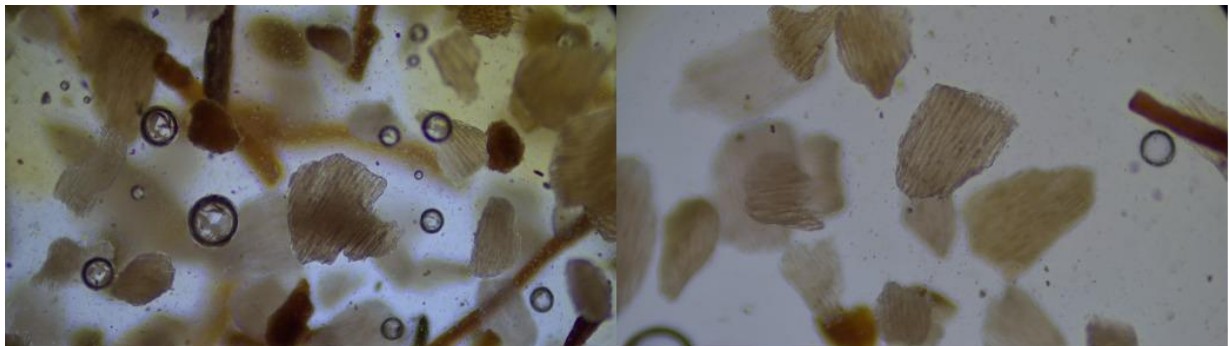

**Figure 4.** Stereo microscope images of sinami seed powder (250 μm) used as an abrasive agent in exfoliant gel.

Then, we visually inspected the abrasive particles in different substrata (water, water/acrylate C10–30, water/sodium laureth sulfate, vegetable oil, and propanol). As expected, the sinami seeds caused the media to change color. Initially, the mixtures were colorless but gradually assumed various shades of reddish-brown (Figure 5). This does not occur with commercial exfoliants because they are pretreated to remove any trace of organic matter. Therefore, it appears that certain dry pulp compounds are attached to the seed within retained fibrous tissue.

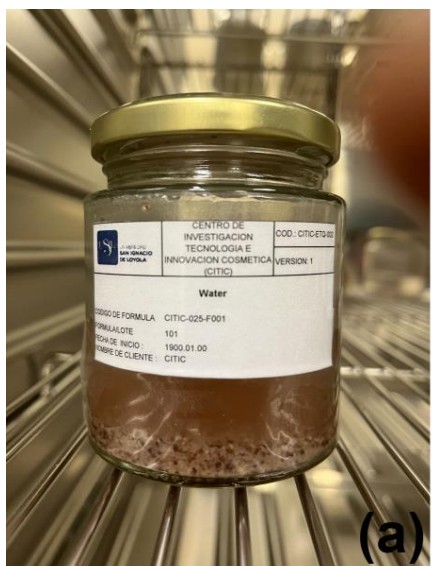
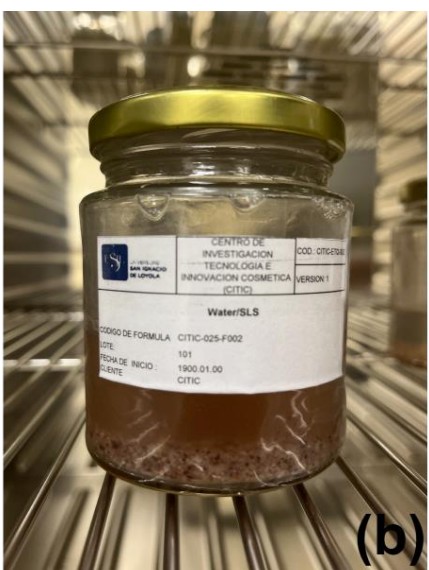
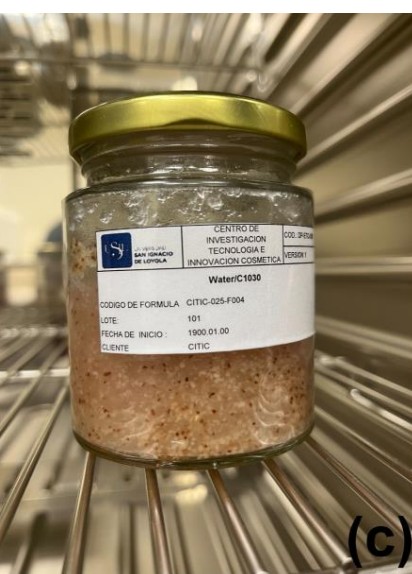

**Figure 5.** Preliminary tests of sinami seed powder immersed in different substrata. (**a**) Water; (**b**) water/sodium laureth sulfate (SLS); (**c**) water/acrylate C10–30 (C1030).

Based on the previous information, a dark in color cosmetic product exfoliant should be formulated. Exfoliants remove old cells to uncover a smooth and rejuvenated skin surface [21]. Cosmetic exfoliants remove impurities through surfactants and abrasives. We evaluated different surfactants to improve the product's functioning and avoid potential incompatibilities with sinami seed powder. Table 5 lists the prototypes' ingredients and their proportions.

**Table 5.** List of ingredients and parameters evaluated in the different prototypes.

| Ingredients/Parameters | Prototype 1 | Prototype 2 | Prototype 3 | Prototype 4 |
|---|---|---|---|---|
| Water | 86.80 | 66.25 | 76.70 | 81.97 |
| Acrylates/C10–30 | 0.40 | 0.45 | 1.00 | 1.00 |
| Sodium cocoyl glycinate | 4.00 | 4.00 | - | - |
| Sodium laureth sulfate | - | 5.00 | 5.00 | 7.00 |
| Sodium lauryl sulfoacetate | - | 5.00 | 2.50 | - |
| Cocamidopropyl betaine | - | - | - | 1.50 |
| Glycerin | 2.00 | 2.00 | 2.00 | 2.00 |
| Methylchloroisothiazolinone | 0.10 | 0.10 | 0.10 | 0.10 |
| Triethanolamine | 0.40 | 0.40 | 0.80 | 0.80 |
| Sinami seed powder | 5.00 | 7.00 | 10.00 | 5.00 |
| Color visualization | White/milky | White/milky | White/milky | Transparent |
| pH value | 4.55 | 6.29 | 6.48 | 5.25 |
| Viscosity (mPa.s) | 890 | 920 | 1240 | 1050 |

In prototype 1, sodium cocoyl glycinate (Hostapon Sci-85p) is the only surfactant. The prototype was unaffected by the presence of foam during processing. However, the detergent action was unexpected, and the abrasive particles could not be suspended. This type of cosmetic requires a gel-type viscosity (Figure 6a).

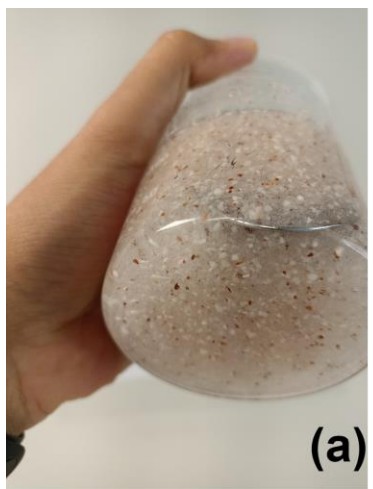 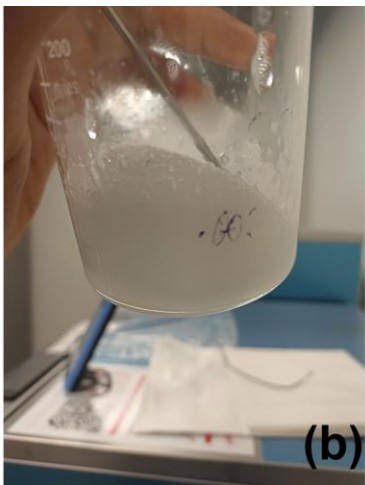 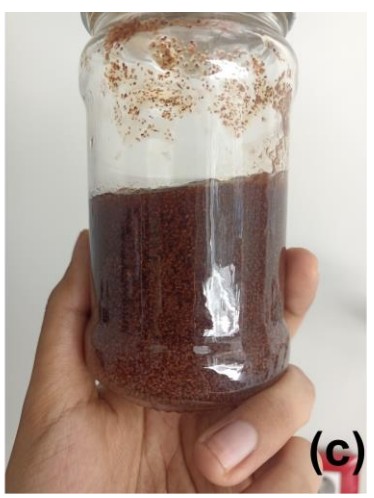

**Figure 6.** Complications (foam formation and opacity) while formulating the sinami seed powder exfoliant. (**a**) Sedimentation of prototype 1; (**b**) foaming of prototype 3; (**c**) presentation of prototype 4.

To improve the cleaning action, we added sodium laureth sulfate (Alkopon) and sodium lauryl sulfoacetate (Lathanol) to prototype 2. However, the bubbles' formation and opacity continued (Figure 6b). Continuing with the trials, Hostapon Sci-85p was eliminated and the concentration of Lathanol was reduced in prototype 3; however, the subsequent inclusion of these surfactants made it even more difficult to see the exfoliating agent. Then, prototype 4 used Alkopon accompanied by another mild surfactant as cocamidopropyl betaine. The mixture remained crystal clear during the dissolution of both with water.

With good visual characteristics in prototype 4, we adjusted the proportion of thickening agents, noting the point where the prototype's abrasives were suspended (Figure 6c). Finally, a commercial exfoliant gel was used to adjust the proportion of the abrasive agent. Figure 7 shows our prototype compared to the commercial gel after the centrifugation test.

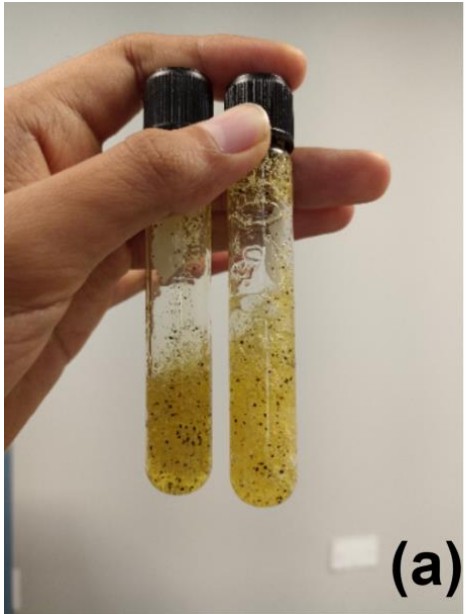
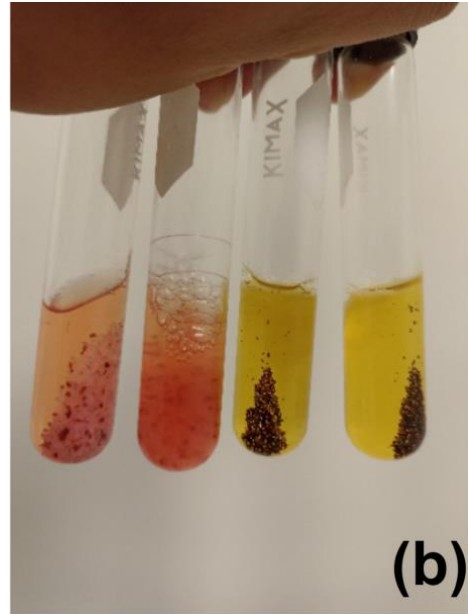

**Figure 7.** Visual comparison of the amount of exfoliating agent in the prototype and the benchmark product on the market. (**a**) The benchmark product before centrifugation. (**b**) The benchmark and prototype after centrifugation.

Most products designed to clean the body should have a pH below 5.0 [22]. In the case of carbomer-based gels, the required pH value is obtained by neutralizing them with alkaline components, which in turn thicken the product. To obtain the highest viscosity, according to the Carbopol Ultrez 20 technical data sheet, the pH must be in the range of 4–9 [23]. In this case, triethanolamine is generally used as a neutralizing agent. However, having a neutral or alkaline product risks microbial growth. During the tests, it was determined that when the sinami seed powder was added, the pH of the medium dropped to 3.73, similar to what was found by Linares-Devia et al. [24] in their formulation of exfoliants based on *passiflora* seeds. The authors observed that when they incorporated the abrasive agent, the pH of the solutions dropped. This effect was attributed to the seeds' carboxylic acid content. It is worth highlighting that their samples were not pretreated as ours were. Based on this information, all the prototypes' pH levels were adjusted to 5.25, where we observed good medium gelation.

### 3.4. Storage Stability Test

Based on previous information, we developed the storage stability test with prototype 4. The evaluation of viscosity determines whether a cosmetic has the right consistency and whether its behavior is stable over time. Figure 8 shows the viscosity readings recorded up to day 84 (3 months). The average viscosity ranged between 700 and 800 mPa·s at room temperature (20 °C) and between 500 and 600 mPa·s under stress conditions (40 °C). Viscosity decreased with increasing temperature. In general, the prototype's viscosity decreased by 35.1% on day 3, 41.4% on day 4, and 44.6% on day 28, compared to the initial values at 40 °C. The prototype was determined to be stable on days 5–21. During this time, its viscosity decreased by only 2.6%. Finally, the viscosity was significantly reduced at the end of the tests (day 84) to 53.2% of its initial value.

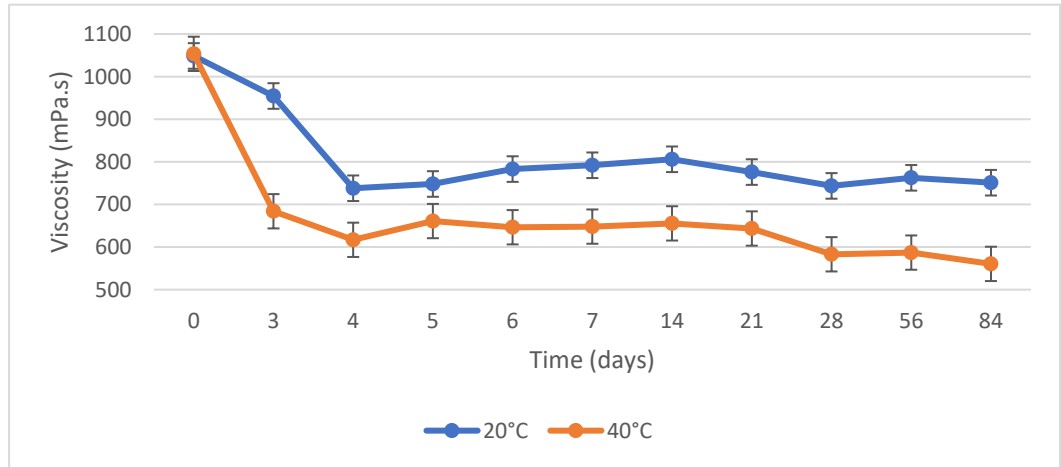

**Figure 8.** Viscosity profile of the exfoliant gel under normal and stressful conditions on storage days 0–84.

Less drastic changes were seen at room temperature, where the viscosity decreased by 8.9% on day 3 and 29.6% on day 4 compared to the initial values (day 0). From day 5, the viscosity remained stable, with an average variation of 20%. The prototype's viscosity decreased by 28.4% between days 0 and 84.

The pH readings recorded up to day 84 (three months) are shown in Figure 9. The initial pH value of the exfoliant gel prototype was 5.25, consistent with Blaak and Staib's recommendations [22]. At room temperature, the prototype's pH increased by 1.52% on day 4 and 0.95% on day 28. An increase of only 0.96% was seen at the end of the stability test. In addition, minor changes were seen in the sample under stress conditions, and at the end of the test, the pH value increased by 0.57%.

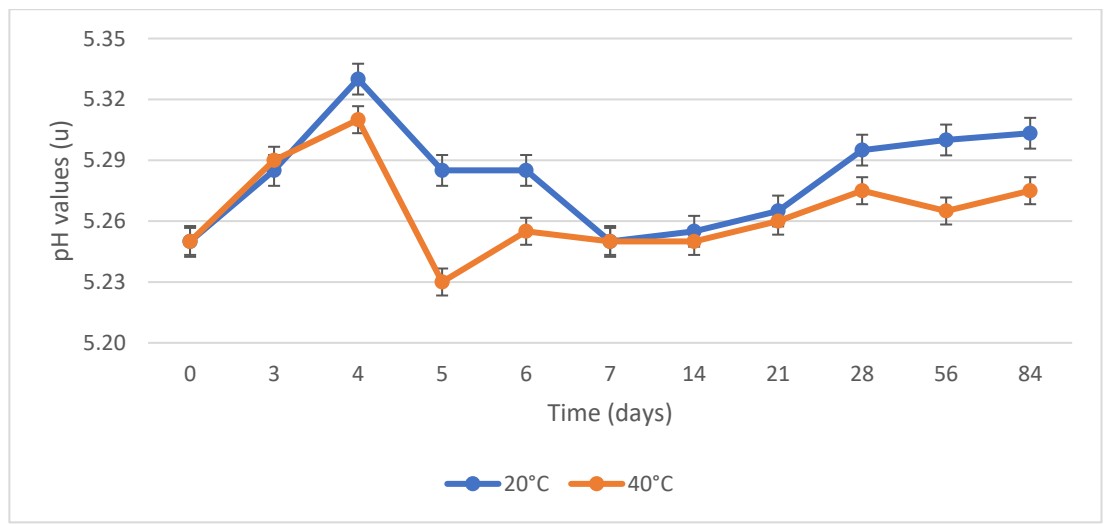

**Figure 9.** pH values of the exfoliant gel under normal and stressful conditions from day 0–84 of storage.

## 4. Discussion

### 4.1. Characterization of the Sinami Seed

The results of the proximal analysis, humidity, and total fiber of the sinami seed powder are summarized in Table 3. The sinami seed had more than triple (25.1%) the moisture content of the açaí seed; however, both seeds contain less moisture than other tropical fruits such as aguaje [25]. This finding holds implications for using sinami as an exfoliating agent. Linares-Devia et al. [24] found that some samples with a high water content retained their

structure, potentially withstanding high compression forces. Therefore, sinami's exfoliating effects will depend on the user's force. Furthermore, hardness is important considering the necessary characteristics of an abrasive particle [24,26].

The lipid levels did not exceed 3% in either seed. The *O. mapora* variety had lipid levels <1%. Although the different varieties of *Oenocarpus* have higher lipid contents, they are mainly distributed in the mesocarp. The pulp of *O. bataua* can reach values between 41.8% and 51.6% of its dry weight [27,28] and the lipid content reaches 35.6% in *O. bacaba* [29].

The protein value of sinami (3.43%) was lower than the reported average value for açaí at 4.89% [18], while values of aguaje are reportedly 8.6%–12.4% [30]. Given its low protein content, sinami is not considered a potential ingredient for a farm animal diet. According to Surayah Osman et al. [31], pets such as hamsters and rabbits require a protein demand of around 12%–25%, far higher than what sinami is able to provide.

The sinami seed features an ash content (1.0%) similar to açaí (1.3%) and aguaje (1.2–1.3%) [18,30]. Linares-Devia et al. [24] recommend caution with these values because they can lead to problems with solution stability. If the ashes are extracted in the aqueous phase it would affect their ionic nature. We report our formulation and stability results in an upcoming section.

The total fiber content of sinami seeds was 9.94%, markedly lower than other Amazonian fruits such as açaí (77.20%) and arazá (36.59%, *Eugenia stipitata*) [32,33]. However, Melo et al. [18] reported that the total fiber values may vary due to the conditioning of the seed or the analysis method used. According to Muñoz et al. [11], sinami contains fibrous tissue that adheres to its external walls making extraction difficult. This fibrous tissue is similar to coconut coir, hairs surrounding the endosperm. Unfortunately, no comparable research was found in the *Oenocarpus* seed research literature.

### 4.2. Characterization of Total Phenolic Content and Antioxidant Activity

Phenolic compounds are a group of micronutrients in the plant kingdom and are an important part of the human and animal diet. They constitute a wide group of chemical substances considered secondary metabolites of plants [34]. These compounds are a focus of increasing interest due to their various biological activities that benefit human health by preventing cancer, cardiovascular disease, and other inflammatory pathologies [35,36].

Table 6 shows the total phenolic content of the sinami seed and other Amazonian fruit seeds found in the literature. These results vary from 12.3 to 452.8 mg GAE/g across different extracts. Sinami seeds demonstrated the lowest levels of total phenolic compounds (12.3 mg/g). The açaí fruit, often used as a reference due to its similarities, had triple the phenol content of sinami (65 mg/g) [18].

**Table 6.** Compilation of works related to the extraction of phenolic compounds in seeds of Amazonian fruits.

| Sample | Common Name | TPC (mg GAE/g) | Reference |
|---|---|---|---|
| *Oenocarpus mapora* | Sinami | 12.3 ± 0.25 | - |
| *Oenocarpus bataua* | Ungurahui | 452.8 | [37] |
| *Euterpe oleracea* Mart. | Açaí | 64.6 | [18] |
| *Artocarpus heterophyllus* | Jackfruit | 27.7 | [38] |
| *Mauritia flexuosa* | Aguaje | 65.5 | [39] |
| *Ambelania duckei* | - | 374.9 | [40] |

The total phenol content in *O. bataua* reached 452.8 mg GAE/g in one study [37]. These results are markedly different from *O. mapora*. Fidelis et al. [41] explained that external factors such as the growing region, climatic conditions, type of soil, and extraction solvent can affect the endogenous synthesis of polyphenols in fruits. Our sample was taken from the Madre de Dios region (Perú), while the *O. bataua* variety was harvested in Pucallpa

(Perú). Future work could focus on sinami seed phenols relative to the area where they were harvested.

Antioxidants are natural or synthetic substances that can prevent or reduce the action of reactive species by inhibiting free radicals and metal complexation [42]. Different assays must be evaluated because each antioxidant substance acts at different stages of the oxidative process [40].

The DPPH assay involves the transfer of electrons and hydrogen atoms and antioxidant activity is measured colorimetrically [43]. The DPPH reagent is deep violet in solution and turns colorless to pale yellow when neutralized by radical scavengers [44]. As reported by Amrani et al. [45], the lower an extract's $IC_{50}$ value, the greater its antioxidant power.

The antioxidant activity of the sinami sample ($IC_{50}$) was 0.34 µg/mL and this is a more effective value when compared to that of other Amazonian fruits such as the piquiá seed (*Caryocar villosum*), pequí pulp (*Caroycar brasilense*), or araçá-boi fruit (*Eugenia stipitata*); here, the DPPH method showed $IC_{50}$ values of 41.07 µg/mL, 9.4 µg/mL, and 0.69 µg/mL, respectively [46–48].

The ABTS assay is also used to measure antioxidant activity. As with the DPPH method, it involves the transfer of electrons and hydrogen atoms; however, ABTS acts in both lipophilic *and* hydrophilic environments and is thus more suitable for quantifying antioxidant activity [40,49].

The antioxidant activity observed with the ABTS method in the seed ($IC_{50}$ = 0.34 µg/mL) surpassed that of the pulp (191.03 µg/mL) and skin ($IC_{50}$ = 43.67 µg/mL) of *Ambelania duckei*, another tropical fruit [40]. Once again, the sinami sample produced high levels of antioxidant activities. Likewise, our sample showed better results than other fruits of tropical plants, such as pitomba (77.3 mg/mL, *Talisia esculenta*) and tucumã-do-Amazonas (*Astrocaryum aculeatum*) [50,51].

Sinami seed extracts have a significant level of antioxidant activity. Wani et al. [52] reported that the lower the $IC_{50}$ value, the greater the antioxidant capacity to reduce free radicals, and powerful antioxidants have $IC_{50}$ values of 0.05–0.10 mg/mL. Therefore, vitamin C, whose strong antioxidant power is reflected in an $IC_{50}$ of 0.009 mg/mL, can be used as a standard referent. Thus, the sinami seed could be repurposed for use in the food or cosmetic industry, as Ribeiro et al. [53] suggested for camu-camu (*Myrciaria dubia*) seeds. However, food by-products that are non-reusable—including residues from other tropical fruits such as acerola (*Malpighia emarginata*), pineapple (*Ananas comosus*), or passion fruit (*Passiflora edulis*)—remain an issue. Moreover, it is difficult to compare published DPPH and ABTS results because of a dearth of scientific investigations into *Oenocarpus* seeds.

*4.3. Exfoliant Prototype Characterization*

The sizes of the most widely used abrasive agents are 250–500 µm, according to Azconia [20]. This size would be ideal for scrubs because it does not cause hypersensitivity reactions or skin irritation. However, the sinami seed is a grainy particle of natural origin; consequently, its size is non-uniform and it has many edges on its borders (Figure 4). Considering these factors, we settled on a particle size of 250 µm.

During the preliminary test, we found the sinami seeds caused the media to change color. Initially, the mixtures were colorless but gradually assumed various shades of reddish-brown (Figure 5). This does not occur with commercial exfoliants because they are pretreated to remove any trace of organic matter. Therefore, it appears that certain dry pulp compounds are attached to the seed within retained fibrous tissue.

Linares-Devia et al. [24] found that dry pulp remnants that were attached to the seed could serve as a chemical exfoliant or antioxidant. However, the darkening could be perceived negatively within the final cosmetic product. Therefore, the authors added dyes to darken the product's tint.

With this information, sodium cocoyl glycinate (Hostapon Sci-85p) is the only surfactant in prototype 1. The prototype was unaffected by the presence of foam during processing. However, the detergent action was unexpected, and the abrasive particles

could not be suspended. This type of cosmetic requires a gel-type viscosity (Figure 6a). Our results suggested that the concentration of gelling agent (acrylates) was inadequate since the abrasive particles could not be suspended.

To complement the cleaning action, we added sodium laureth sulfate (Alkopon)—a surfactant widely used in cosmetics. Unfortunately, some users find sodium laureth sulfate irritating to the skin. In such cases, a milder surfactant such as sodium lauryl sulfoacetate (Lathanol) is typically added. Both ingredients are excellent foaming agents and detergents. Both of these qualities enhance the distribution of the cosmetic following skin application. However, two issues arose: bubbles and solution color changes.

As shown in Figure 6b, prototypes 2 and 3 produced lots of foam during testing. As expected, the aqueous surfactant solutions were prone to foaming. Kelleppan et al. [54] explain that the molecules of the surfactants are used to orient their hydrophilic head groups toward the solution with their hydrophobic tail groups pointing toward the air, thus allowing appreciable volumes of foam to develop and remain stable over time. Even though foam is a desirable property in a cleaning product—because it drags the hydrophobic particles from the substrate and improves the consumer's perception of the product's efficacy [55]—it is aesthetically unfavorable since the foam conceals the mixture's sinami particles. Furthermore, use of both surfactants changed the mixture's color. The mixture remained crystal clear during the dissolution of water with Alkopon; however, the subsequent inclusion of Lathanol changed the color of the medium to a milky one, making it even more difficult to see the exfoliating agent.

Because the inclusion of two surfactants was unsuccessful, we decided to continue with sodium laureth sulfate (Alkopon) due to its lower cost in prototype 4 [56]. However, we included cocamidopropyl betaine to counteract potential irritation. This cosmetic ingredient is a surfactant, which is mild and skin-friendly, used in conditioners, body washes, and other personal care products. Many prefer cocamidopropyl betaine over other surfactants due to its synergistic effects with other cosmetic ingredients, thus favoring the formulation of products [57,58]. We adjusted the proportion of thickening agents to prepare prototype 4, noting the point where the prototype's abrasives were suspended (Figure 6c). Finally, a commercial exfoliant gel was used to adjust the proportion of the abrasive agent. Figure 7 shows our prototype compared to the commercial gel after the centrifugation test.

### 4.4. Storage Stability Test

The stability of a cosmetic product involves physical, chemical, and microbiological analyses. Physical stability is measured by the absence of phase separation and the modification of the product's rheological characteristics [59]. We evaluated the stability of our prototype sinami exfoliant gel by comparing its pH and viscosity at room temperature and under stress conditions.

The stress condition accelerates the expected changes due to normal storage and use conditions. Despite the heat treatment, no significant differences in color or particle suspension were observed in the prototype; however, a thin layer of water was observed on the exfoliant gel samples during the stability tests. According to Zięba et al. [60], this sheet of water occurs when water evaporates from the product due to increased heat; however, when the solution was homogenized with a palette, it dispersed without major complications.

Viscosity refers to how a cosmetic product spreads across human skin [61]. The viscosity readings recorded up to day 84 (3 months) ranged between 700 and 800 mPa·s at room temperature (20 °C) and between 500 and 600 mPa·s under stress conditions (40 °C). Notably, our results differ from Zięba et al. [60] who examined a shower gel formulation and found the dynamic viscosity to increase by 5% after three months of storage. In contrast, in our study, the dynamic viscosity was reduced by 28.4%. Although the data obtained showed the viscosity below that of commercial gel exfoliants (3600–6000 mPa·s), future sensory evaluation tests could improve this formulation.

The variation in pH in a cosmetic product can represent instability, direct or indirect contamination during formulation, or possible chemical reactions between raw materials [62,63]. Human skin's pH ranges from 4.1 to 5.8, depending on location. Thus, products that are applied to skin should be formulated in this range.

According to Berthele et al. [64], these changes in pH are probably due to yeast and mold growth. The initial pH value of the exfoliant gel prototype was 5.25, consistent with Blaak and Staib's recommendations [22], and at the end of the test, the pH value increased by 0.57%. The prototype's pH varied minimally and remained within an acceptable limit at 0.06% after a preservative (Prodice CG) was added to the base formula. These results are encouraging because they could indicate that this formulation may be adequate for cosmetic use. Importantly, cosmetic products must maintain physical stability throughout their useful life so that the user does not perceive changes as the product is repeatedly applied over time.

## 5. Conclusions

These results fill a knowledge gap related to the sinami (*O. mapora*) seed. We found the sinami seed to have a high moisture content (25.1%) compared to other Amazon seeds, but low crude fiber (9.95%) and total lipid (0.44%) content. We noted that most of these elements came from the pulp and peel. Sinami extracts showed high antioxidant activity in vitro against DPPH $IC_{50}$ (0.34 mg/mL) and ABTS $IC_{50}$ (0.10 mg/mL) radicals. A prototype with sodium laureth sulfate and cocamidopropyl betaine creates a crystal gel that is compatible with sinami seed powder. These results suggest that sinami seed powder holds promise as a commercial exfoliant. The behavior of the prototype across all studied variables (pH and viscosity variation) affirms its stability. However, further studies at the exfoliation level (hardness and morphology) are needed, in addition to chemical and microbiological stability tests, to verify its efficiency.

**Author Contributions:** Conceptualization, F.L.R.-O.; methodology, F.L.R.-O., J.H., P.L. and F.R.-E.; investigation, F.L.R.-O. and F.R.-E.; writing—original draft preparation, F.L.R.-O. and F.R.-E.; writing—review and editing, F.R.-E. and A.M.M.; visualization, J.H., A.M.M. and F.R.-E.; supervision, P.L. and A.M.M. All authors have read and agreed to the published version of the manuscript.

**Funding:** This research received no external funding.

**Institutional Review Board Statement:** Not applicable.

**Informed Consent Statement:** Not applicable.

**Data Availability Statement:** Not applicable.

**Conflicts of Interest:** The authors declare no conflict of interest.

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
