# Peer review of "Development and Functionality of Sinami (Oenocarpus mapora) Seed Powder as a Biobased Ingredient for the Production of Cosmetic Products"

_cosmetics, doi:10.3390/cosmetics10030090_

Round 1

Reviewer 1 Report

In the MS entitled "Development and functionality of sinami (Oenocarpus mapora) seed powder as a bio-based ingredient for the production of 3 cosmetic products," the authors had an interesting purpose. They determined the phytochemical profile of O. mapora seeds and their antioxidant activity. Then, they manufactured an exfoliating cosmetic product, studied under normal and stressed 26 conditions (40°C) for 3 months, maintaining a pH value of 5.25 and a final viscosity of 700-800 mPa.s 27 and 600-500 mPa.s in normal and stress condition, respectively. 

The reviewer's comments are available below, analyzing each section:

Abstract

In the abstract, the findings regarding the exfoliant gel are missing.

Introduction

This section could be clearer and could contain more details regarding the physicochemical profile and applications of studied species.

Material and Methods

This section contains the following subsections: 

2.1. Chemical products

2.2. Sampling and treatment of Oenocarpus mapora samples - the authors are encouraged to check and correct this subtitle, avoiding repetition

2.3. Physicochemical characterization of the sinami seed - is it only one seed? Please, check and correct.

2.3.1. Proximal composition

2.3.2. Extraction process

2.3.3. Determination of total polyphenol content

2.3.4. DPPH radical scavenging activity

ABTS radical scavenging activity (without number; the reviewer suggests changing the name of this subsection to 2.4. Antioxidant activity, with 2.4.1. DPPH free radical scavenging activity and 2.4.2. ABTS free radical scavenging activity.

2.3.5. Sieve separation - the reviewer thinks it is not a part of Physicochemical characterization; moreover, the authors should clarify if this very fine powder is used to determine phenolic compounds and antioxidant activity or if it is obtained for exfoliant formulation manufacturing.

2.4. Exfoliant formulation

2.4.1. pH and texture

2.4.2. Storage stability test 

Other comments:

1. Lines 128 and 129: Determination of total phenolic content is better, as in Results. Please, check and correct.

2. Line 130: the authors should add the reference number after Ferrer Cutire et al., not (Ferrer Cutire et al., 2022). Please, check and rectify.

3. Lines 143 and 149: IC50 is better than IC50. Please, rigorously check and correct the entire MS text.

4. Lines 165,169, and 345: The authors are encouraged to show the Tables 1, 2, and 6 captions, as in Tables 3-5.

5. Line 170: "Half" after ":" with "h" is better; please, check and correct.

Results and Discussion

In its current form, this section is structured as follows:

3.1. Characterization of the sinami seed

3.2. Characterization of phenolic compounds and antioxidant activity

3.3. Exfoliant prototype characterization - the authors are encouraged to be clearer about all prototypes: what is the reason for using them, and if the final tests were performed using all of them or only one or two. They appear only in Table 6; from the following MS text and Figures 6 and 7, it can not be very clear if all 4 prototypes are available. The authors are encouraged to show more clearly what is each prototype's utility in their research.

3.4. Storage stability test

 1. The authors are encouraged to separate the Results from the Discussion section.

Their study is vast enough to put all results in a single section and conduct suitable discussions. 

In the current version, mixed with discussions and observations from other studies, identifying the authors' contribution is hard for the reader.

2. Table 4, for example, could be included in the Discussion because it is entitled  "Compilation of works related to the extraction of phenolic compounds in seeds of Amazonian fruits" and contains different data from the literature. 

3. Line 207, Table 3: the scientific names of plants must be written in italics; please, rigorously check and correct the entire MS text.

4. Line 261: total phenolic content is better; please, check and correct.

5. Line 293: Please check and correct the text of the figure caption.

After performing all requested changes, the Conclusions could be substantially and clearly supported by both independent sections, Results, and Discussion.

References:

The authors are encouraged to check and edit all references according to MDPI instructions for authors.

English language editing and style are unwieldy; these aspects must be improved to avoid all misprints and danglings. Maybe, in the revised form, the MS data should be better organized, and all information should be more explicit for the reader. 

Author Response

Response 1: The translation of the manuscrit was improved to define properly some terms and re-orgnized ideas. Abstract and Introduction was reduced. In case of Materail and Methods, the subsections was modified according to reviewer 1 suggestions. Mistakes about scientific names and tables with missing names was solved.

Response 2: Results and Discussion section is re-organized and follow the same struction.

Reviewer 2 Report

This is a poorly written manuscript on possibly an interesting study. The authors should use a scientific style and proofread the English language before they resubmit this article. Latin names are always italic for example. Please find appropriate guidelines for reporting and add missing information, to the abstract but also to the manuscript, such as the limitation section. I do not find figures to be appropriate for a scientific article, could you make figures in program and add originals to the supplement?

Hard to follow, please improve

Author Response

Response 1: The translation of the manuscrit was improved to define properly some terms and re-orgnized ideas.

Round 2

Reviewer 1 Report

The reviewer appreciates the authors' substantial efforts to rectify their manuscript according to the first review report. 

The manuscript is significantly improved.

Of the total references, 28 have been published in the last 5 years. However, the authors are encouraged again to rigorously check and edit all references according to MDPI instructions for authors.

The English Language quality is considerably augmented.

Reviewer 2 Report

From my point of view this manuscript has been improved and it is now suitable for publication